# Attosecond electronic timing with rising edges of photocurrent pulses

Minji Hyun[1], Changmin Ahn[1], Yongjin Na[1], Hayun Chung [2✉] & Jungwon Kim [1✉]

There has been remarkable progress in generating ultralow-noise microwaves from optical frequency combs in the last decade. While a combination of techniques has enabled tens to hundreds of attoseconds residual jitter in microwave extraction, so far most of research efforts have been focused on extracting single-tone microwaves from combs; there has been no study on the noise properties of photocurrent pulses directly extracted from the photo-diode. Here, we reveal that the residual jitter between optical pulses and rising edges of photocurrent pulses can be in the tens of attoseconds regime. The rising-edge jitter is much lower than the falling-edge jitter, and further, this ultralow rising-edge jitter could be obtained by both p-i-n and (modified-)uni-travelling-carrier photodiodes. This finding can be directly used for various edge-sensitive timing applications, and further shows the potential for ultrahigh-precision timing using silicon-photonic-integrable on-chip p-i-n photodiodes.

[1] School of Mechanical and Aerospace Engineering, Korea Advanced Institute of Science and Technology (KAIST), Daejeon 34141, Korea. [2] Department of Electronics and Information Engineering, Korea University, Sejong 30019, Korea. ✉email: hcchung@korea.ac.kr; jungwon.kim@kaist.ac.kr

Precise optical timing is possible by exploiting the ultralow-phase noise repetition rates of optical pulse trains generated from femtosecond mode-locked lasers[1,2] or microresonator-based soliton frequency combs[3,4]. On the one hand, femtosecond mode-locked lasers and frequency combs have intrinsically low timing jitter. High-performance, free-running solid-state or fibre mode-locked lasers show sub-100-attosecond (as) timing jitter in the optical pulse trains on the fast time scales (e.g., over ∼1 ms)[5–7]. On the other hand, using the optical frequency division (OFD), i.e., locking the mode-locked frequency comb to a cavity-stabilized continuous-wave laser, sub-femtosecond-level optical timing of frequency combs can be maintained over longer time scales, e.g., ∼1 s time scale, as well[8,9]. Using such attosecond optical timing properties, ultralow-phase-noise, single-tone microwave signals can be extracted by high-speed photodetection followed by microwave amplification and filtering of a harmonic frequency component of repetition rate ($nf_{rep}$, where $f_{rep}$ is the fundamental repetition rate and $n$ is an integer). However, transferring the precise optical timing to the electronic domain has turned out to be highly non-trivial due to the excess phase noise introduced in the optical-to-electronic (OE) conversion processes[10–13] in high-speed photodiodes. Recently, there has been rapid progress in overcoming excess phase noise problems, where effective approaches include repetition-rate multiplication[11,14], design of higher linearity photodiode structures[11,15,16], including modified-uni-travelling-carrier (MUTC) photodiodes, and finding the optimal photodiode operation conditions for achieving minimal amplitude-to-phase conversion (APC) coefficients[11,13,17,18]. By combining these approaches, recently, 68-as-level timing in OE conversion in 10-GHz microwave extraction has been demonstrated[19].

In fact, upon photodetection, an optical pulse train is first converted to the photocurrent pulse train. Although there have been various studies in reducing excess phase noise in the OE conversion, they have been focused on the excess phase noise between the optical pulse train and the sinusoidal microwave at a specific harmonic frequency, $nf_{rep}$[9,19,20]. So far, to our knowledge, there has been no study on the excess noise in the very first stage of OE conversion process, i.e., from the optical pulse train to the electrical pulse train. Note that, as some relevant researches on photocurrent pulse properties, previous studies on the input optical power-dependent photocurrent pulse shape showed much stronger dependence on the falling edge than the rising edge[13,21–23], which suggests different behaviour in excess noise between rising- and falling edges of photocurrent pulses. There were also previous studies on elucidating ultrafast photocharge transports in high-speed photodiodes using electro-optic sampling techniques[24–26], showing strong dependence on the bias voltages (electric field) and optical pulse energy.

Here, using an electro-optic sampling-based ultrasensitive timing detection method[27,28], we characterize the excess timing jitter between the optical pulse train and the photocurrent pulse train with 51-as (r.m.s.) resolution over 1 MHz bandwidth. Our study reveals that the residual timing jitter between optical pulses and rising edges of photocurrent pulses can reach 64 as (r.m.s.) over 1 MHz bandwidth, when the measurement noise floor contributes 51 as. We found that the rising-edge jitter (in the range of tens–hundreds as) is much lower than the falling-edge jitter (in the range of few–tens fs), and further, this attosecond rising-edge jitter can be obtained from both p–i–n and MUTC photodiodes. To fully exploit only the attosecond rising-edge jitter of photocurrent pulses, we also demonstrate the use of photocurrent pulse shaping with balanced photodetection. Our findings and demonstrated methods can be used for various edge-sensitive, ultralow-jitter timing applications, including analogue-to-digital converters[29–31], ultrahigh-speed data links[32,33], 5G

communications[34], on-chip clock distribution networks[35–37] and frequency synthesizers[38], using CMOS-compatible p–i–n photodiodes[39,40] that can be also integrated on silicon photonic chips.

## Results

**Timing jitter measurement concept and setup.** The concept and experimental setup of characterizing residual timing jitter between femtosecond optical pulses and the generated photocurrent pulses is shown in Fig. 1a. As the photocurrent pulse duration is much longer than the optical pulses (e.g., ∼30-ps rise time and ∼250-ps fall time for photocurrent pulses when a 12-GHz p–i–n photodiode is used for detecting ∼200-fs-long optical pulses with ∼13 pJ pulse energy; see Fig. 1b for the measured photocurrent pulse shapes for 12-GHz p–i–n and MUTC photodiodes with different input power levels), we characterized the residual timing jitter by placing the optical pulse at different temporal positions along the photocurrent pulse. An electro-optic sampling-based timing detector (EOS-TD) is used for the jitter characterization (see Methods and refs. [27,28] for more information). An optical pulse train generated from a 250-MHz femtosecond mode-locked Er-fibre laser is split by a coupler and applied to both the photodiode-under test and the EOS-TD. Using a variable optical delay line (VDL in Fig. 1a), the relative temporal position between optical pulses and photocurrent pulses can be tuned. Three different commercially available photodiodes (12-GHz p–i–n photodiode, 12-GHz MUTC photodiode and 22-GHz MUTC photodiode) are used for generating photocurrent pulses (see Methods).

**Timing jitter measurement results.** Figure 1c shows the relative timing jitter power spectral densities (PSDs) at the 50% of peak amplitude positions of the rising and falling edges of photocurrent pulses for 12-GHz p–i–n (20-V bias), 12-GHz MUTC (8-V bias) and 22-GHz MUTC (8-V bias) photodiodes under test. The average photocurrent is set to 3 mA for all cases. The first observation is that the rising-edge jitter can reach ultralow jitter of sub-100-as regime. For the 22-GHz MUTC photodiode case, the r.m.s. integrated timing jitter of 80 as [integration bandwidth: 1 Hz–1 MHz] (curve (iii)) is obtained when the EOS-TD detection noise (curve (vii)) contributes to 51 as. The jitter PSD can reach 50 as Hz$^{-1/2}$ level (−173 dBc/Hz of equivalent phase noise at 10-GHz carrier), which is limited by the detection noise floor of the used EOS-TD (curve (vii)). Since the optical pulse train generated from mode-locked laser combs can have sub-100-attosecond-level absolute timing jitter[1,5–7], the absolute timing jitter of electric pulse rising edges can also reach the sub-femtosecond regime.

Note that the EOS-TD measurement noise floor (curve (vii)) is determined when the optical input to the photodiode-under-test is blocked and no electrical signal is applied to the EOS-TD. We also separately estimated the impact of path length mismatch between the two fibre paths (i.e., one path to the photodiode-under-test, the other path to the optical input of the EOS-TD in Fig. 1a; see Methods), and as shown by curve (viii) in Fig. 1c, it is much lower than the rising-edge jitter measurement results.

The second observation is that the rising-edge jitter PSD is much lower than the falling-edge jitter PSD by up to ∼40 dB (i.e., a factor of ∼100 reduction in timing jitter). For the 12-GHz MUTC (p–i–n) photodiode cases, the r.m.s. rising- and falling-edge jitters [1 Hz–1 MHz] are 140 as (180 as) and 11 fs (12 fs), respectively. The third observation is that the sub-200-attosecond rising-edge jitter can be obtained by both p–i–n and MUTC photodiodes when sufficiently large bias voltages are applied. Note that, previously, for ultralow-noise microwave extraction,

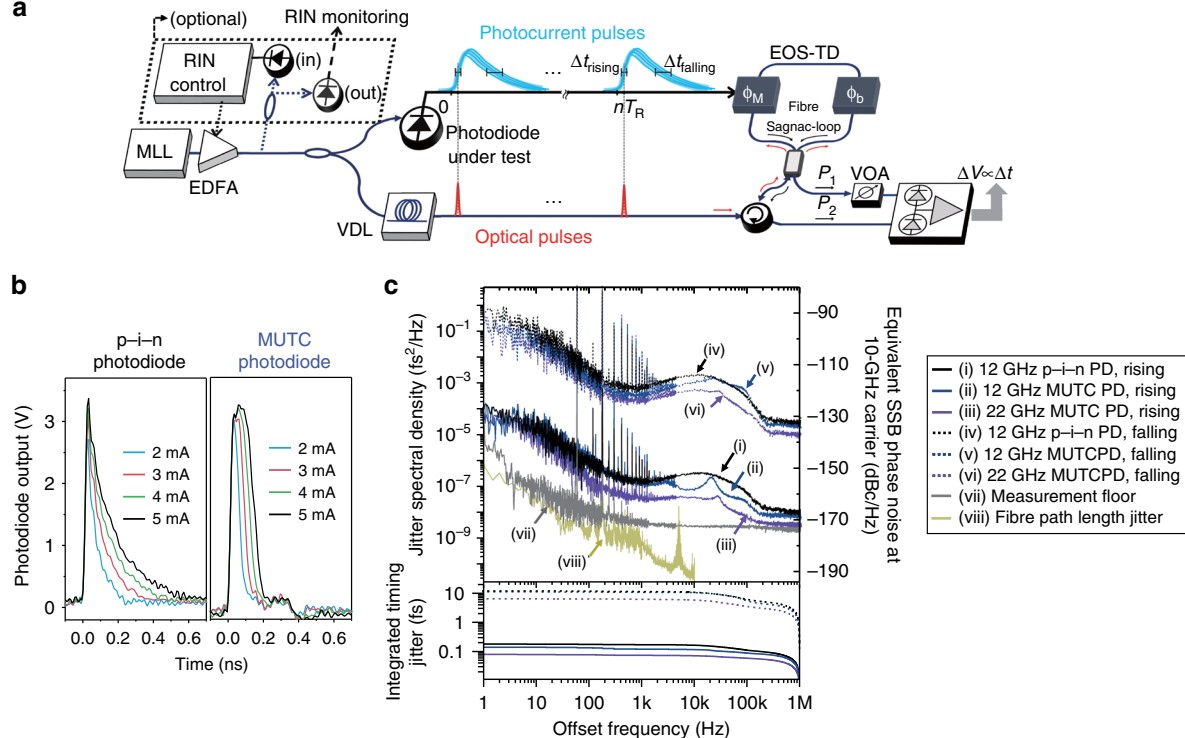

**Fig. 1 Photocurrent pulse timing jitter measurement concepts and results. a** Concept and experimental setup for characterizing relative timing jitter between femtosecond optical pulse train and the generated photocurrent pulse train. MLL mode-locked laser, EDFA erbium-doped fibre amplifier, VDL variable optical delay line, $\phi_M$ unidirectional electro-optic phase modulator, $\phi_b$ $\pi/2$ nonreciprocal phase bias, EOS-TD electro-optic sampling-based timing detector[27,28], VOA variable optical attenuator. Note that RIN control is optional. **b** Photocurrent pulse profiles (voltage across a 25-ohm impedance) measured by a 33-GHz real-time oscilloscope. A 12-GHz p–i–n photodiode (20-V bias) and a 12-GHz MUTC photodiode (8-V bias) for different average photocurrent levels are used. **c** Timing jitter power spectral densities measured by the EOS-TD at the middle of rising edges of (i) 12-GHz p–i–n (black solid), (ii) 12-GHz MUTC (blue solid), (iii) 22-GHz MUTC photodiodes (violet solid) and at the middle of falling edges of (iv) 12-GHz p–i–n (black dashed), (v) 12-GHz MUTC (blue dashed) and (vi) 22-GHz MUTC photodiodes (violet dashed). (vii) EOS-TD measurement background noise for 22-GHz MUTC photodiode case (grey solid). (viii) Estimated fibre path length jitter (dark yellow solid, see Methods). The bottom curves indicate the integrated root-mean-square (r.m.s.) timing jitters, which are (i) 180 as, (ii) 140 as, (iii) 80 as, (iv) 12 fs, (v) 11 fs, (vi) 6.5 fs, when integrated from 1 Hz to 1 MHz offset frequency.

special-structured MUTC[15] or p–i–n[9] photodiodes with high linearity and low flicker noise were used to obtain attosecond-level timing at a specific microwave oscillation. Here, we found that general-purpose p–i–n photodiodes could also achieve attosecond-level timing at the rising edges of photocurrent pulses. As will be shown later with simulation result and relative intensity noise (RIN) controlled experiment result, the amplitude-to-timing conversion (ATC) is the dominant factor for the photocurrent edge jitters. Note that, in the following discussions, we will use the ATC (i.e., timing change per amplitude change) instead of the APC (i.e., phase change per amplitude change) which has been more widely used for the single-tone microwave extraction research[17,41]. Since the photocurrent pulse itself contains many frequency components, the impact of amplitude noise can be quantified more meaningfully by considering the timing change rather than the phase change.

**Parameter dependence of timing jitter.** Figure 2a shows the measured, amplitude-normalized photocurrent pulse shapes from 12-GHz p–i–n and MUTC photodiodes with different input optical power levels and bias voltages. Higher bias voltage and lower input optical power lead to shorter falling tail lengths. Figure 2b shows the impact of bias voltage on integrated timing jitter at different temporal positions (at 50% of peak amplitude for the rising edge and 10, 30, 50, 70 and 90% of peak amplitude

for the falling edge). All measurements were performed at 3-mA average photocurrent. As expected, generally, the jitter is reduced by increasing the bias voltage since higher bias voltage relieves the space-charge-screening effects in a photodiode[15,42]. Note that, only for the rising-edge of MUTC photodiode, the jitter is slightly increased with higher bias voltage because the saturation power is increased due to the higher bias, so that the photodiode is less saturated and the jitter is increased by the ATC. While the falling-edge jitter of the MUTC photodiode has less position dependency, the p–i–n photodiode jitter generally increases toward the end of falling edges. Figure 2c shows the input optical power dependence on timing jitter from 2 to 5 mA average photocurrent with an interval of 0.5 mA for 12-GHz p–i–n and MUTC photodiodes. For high photocurrent conditions where the photodiodes are sufficiently saturated, rising-edge jitter is in the sub-200-as level and nearly unchanged. On the other hand, the falling-edge jitter tends to increase with optical power because the nonlinearity of a photodiode is grown by stronger space-charge-screening effects. This general tendency also appears between the APC coefficient and optical power[18]. Note that we also tried to measure the falling-edge jitter at lower photocurrent levels (e.g., <1 mA) to find the recently studied minimum-width impulse response point[23], where the impulse response and the coupling of amplitude noise to centre-of-mass jitter are minimized. However, due to the limited EOS-TD resolution in lower current regime, the measurements were limited by the EOS-TD resolution itself

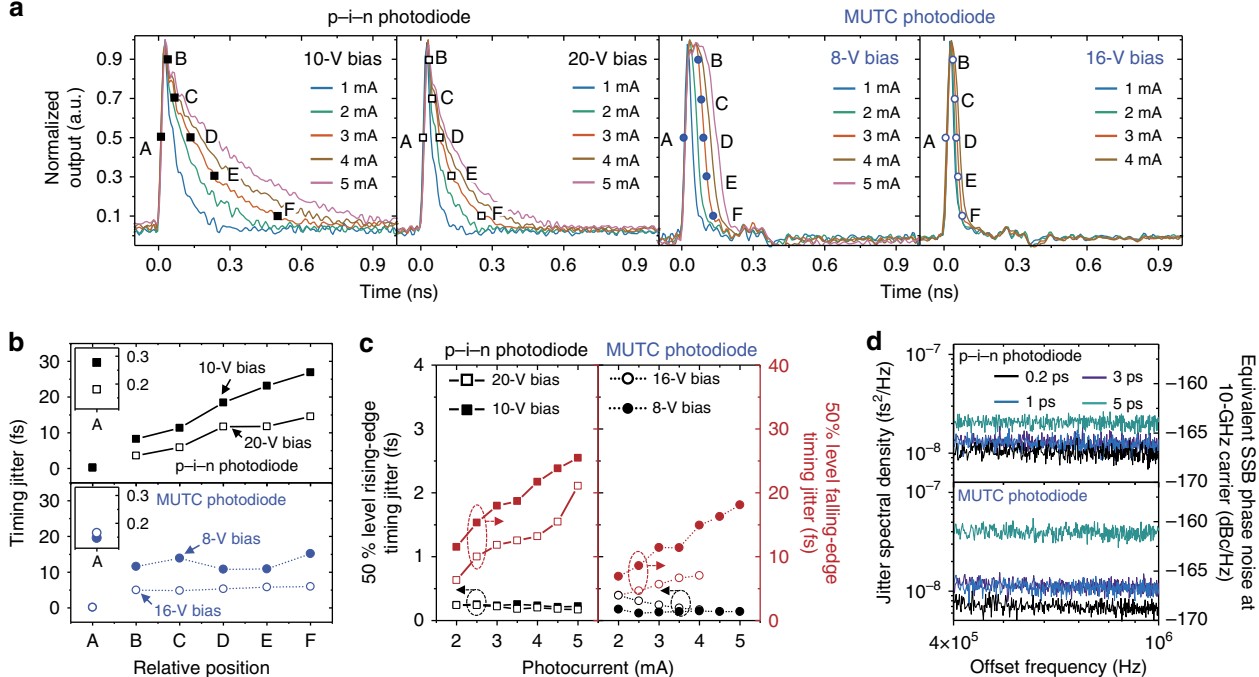

**Fig. 2 Parameter dependence of photocurrent pulse timing jitter. a** Photocurrent pulse shapes of 12-GHz p–i–n and MUTC photodiodes for different input optical power levels and bias voltage conditions. **b** Bias voltage and pulse position dependence on timing jitter of 12-GHz p–i–n and MUTC photodiodes for low (10 V for p–i–n and 8 V for MUTC) and high (20 V for p–i–n and 16 V for MUTC) bias voltages. Both measurements are performed at 3-mA average photocurrent. **c** Input optical power-dependent timing jitter measured at the 50% level of both rising and falling edges. Note that the measurement of 12-GHz MUTC photodiode for 16 V bias voltage is limited up to 4-mA photocurrent due to photodiode thermal failure. **d** Timing jitter PSDs of 12-GHz p–i–n (20 V bias) and MUTC (8 V bias) photodiodes at the middle of rising edge for selected optical pulsewidths (0.2, 1, 2 and 5 ps).

and could not distinguish differences in jitter PSDs. Improvement of EOS-TD resolution and more careful jitter study in such low photocurrent regime will be an interesting future research topic. Finally, we also examined the timing jitter dependence on the input optical pulsewidth in the range of 200 fs to 5 ps. For both types of photodiodes, the timing jitter of falling edge is almost independent on the input pulsewidth. For rising-edge jitter, some degradation with long optical pulsewidth is observed (Fig. 2d) similar to previous result of microwave phase noise[43]. This optical pulsewidth dependence might be a result of dynamic changes in the space-charge-screening effect, which is similar to the previous study on the excess noise from photocarrier scattering[44]. When the optical pulsewidth is shorter, the photocarriers are likely to experience the same space-charge-screening effect, which results in a smaller variation in transit time and lower timing jitter at the rising edge.

**Impact of ATC on timing jitter**. To examine the origin of the pulse-edge-dependent timing jitter of photocurrent pulses, modelling, simulation and additional measurements were conducted. It turns out that the coupling of optical intensity (amplitude) noise to the timing noise is the dominant factor. First, to obtain the theoretical ATC coefficients, we modelled the transfer function of photodiodes with optical pulses and photocurrent pulses as the input and output, respectively. Since the optical pulse duration is much shorter than the photodiode response time, we can regard the measured photocurrent pulse shape as the impulse response of the photodiode-under-test. Hence, the photodiode transfer function can be obtained by the Fourier transform of the measured photocurrent pulse shape. We have recorded the time responses using a 33-GHz bandwidth real-time oscilloscope. Note that the time response of the photodiode is much shorter than the inverse of repetition rate, so the

photodiode has enough time to get back to the equilibrium. For photodiode transfer function modelling, 12-GHz p–i–n and MUTC photodiodes are used. The Fourier transformed spectra of the impulse time responses at the 3-mA average photocurrent level are shown as solid curves in Fig. 3a. The fitted responses of photodiode transfer funciton, which are based on the well-known equivalent-circuit models[45,46] of the p–i–n and MUTC photodiodes, are also shown as dashed curves in Fig. 3a, which shows a fairly good agreement with the Fourier transform of the impulse responses. Using the obtained transfer functions of photodiodes, an ATC coefficient, which represents the timing variation arising from a fractional amplitude fluctuation, can be obtained. The ATC coefficients can be obtained from the linear system simulation with transfer functions (see Methods). For the 12-GHz p–i–n (MUTC) photodiode, the ATC coefficients [in s $(\Delta P/P)^{-1}$ unit] at 3-mA average photocurrent condition were computed as 2.8 ps (1.1 ps) and 110 ps (60 ps) at the middle of rising edge and falling edge, respectively (see Fig. 3b).

Then, the ATC coefficients were experimentally measured by a method similar to the APC coefficients measurement methods[47] (Fig. 3c; see Methods). The acousto-optic modulator (AOM) is used for amplitude modulation and the EOS-TD is used to measure the timing change caused by the amplitude modulation. As shown in Fig. 3b, the measured ATC coefficients [in s $(\Delta P/P)^{-1}$ unit] at 3-mA average photocurrent condition are 3.8 ps (1.7 ps) and 81 ps (84 ps) at rising edge and falling edge, respectively, for a 12-GHz p–i–n (MUTC) photodiode. For both simulation and measurement results, the ATC coefficients are vastly different between rising edge and falling edge.

With the measured laser RIN spectrum (curve (i) of Fig. 4a) and the computed and measured ATC coefficients of photodiodes, the ATC contribution to the timing jitter was predicted. The calculated ATC-originated jitter PSDs, measured ATC-originated jitter PSDs

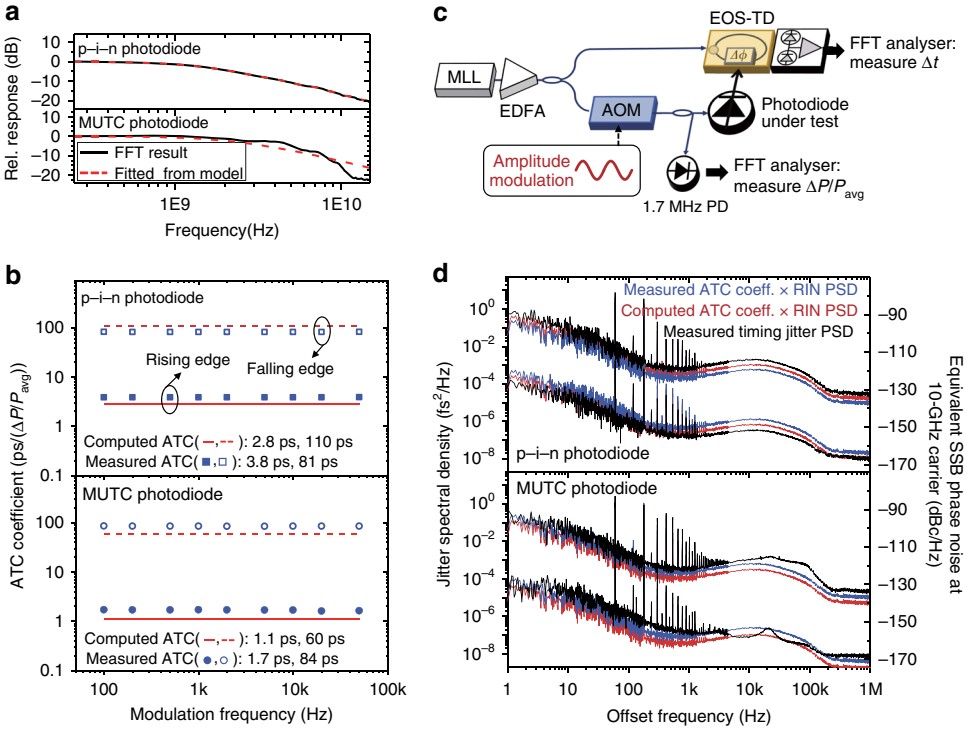

**Fig. 3 Impact of amplitude-to-timing conversion on photocurrent pulse timing jitter. a** Magnitude of Fourier transforms of photocurrent responses of 12-GHz p–i–n and MUTC photodiodes at the 3-mA average photocurrent level (solid curves). The fitted results from photodiode models[45,46] are shown as dashed curves. **b** ATC coefficients measured and calculated at various modulation frequencies at rising edge (solid line) and falling edge (dashed line). **c** Experimental setup for measuring the ATC coefficients. **d** Comparison between measured timing jitter spectra with the computed ATC-originated jitter spectra and the measured ATC-originated jitter spectra, which shows a good agreement for both rising and falling edges of p–i–n and MUTC photodiodes.

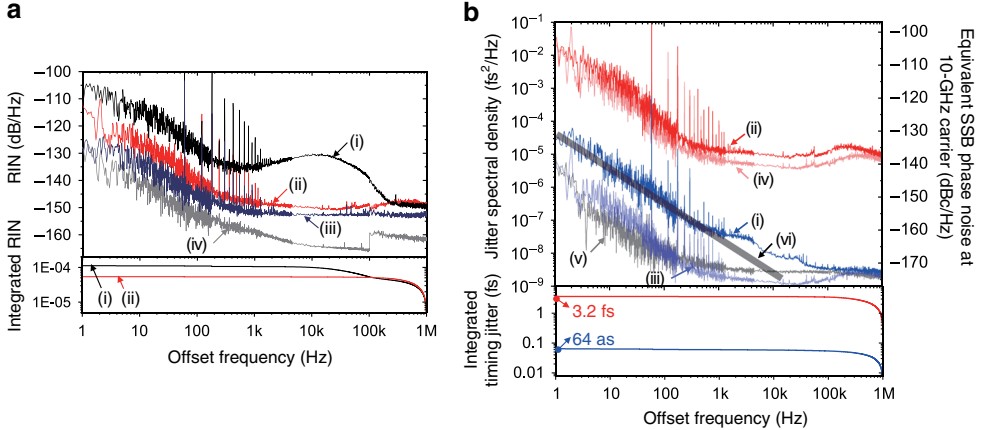

**Fig. 4 Impact of relative intensity noise (RIN) suppression on photocurrent pulse timing jitter. a** RIN suppression result: (i) without and (ii) with RIN suppression. (iii) Equivalent RIN noise floor of the voltage reference used for RIN suppression. (iv) Dark noise floor of the detection system. **b** Measured timing jitter PSDs of the 22-GHz MUTC photodiode with RIN suppression: (i) measured rising-edge jitter PSD, (ii) measured falling-edge jitter PSD, (iii) projected ATC-originated rising-edge jitter contribution, (iv) projected ATC-originated falling-edge contribution, (v) EOS-TD measurement noise floor and (vi) predicted photodiode flicker noise[9,18,19]. The bottom curves indicate the r.m.s. integrated rising-edge and falling-edge timing jitters [1 Hz–1 MHz].

and the measured jitter PSDs are presented and compared in Fig. 3d. The ATC-originated jitter spectra agree fairly well with the measured jitter PSD data. From this result, we can deduce that the ATC noise is the main source of both rising- and falling-edge jitters, which suggests that better RIN control may lead to even lower timing jitter levels.

From this finding, we can also connect a link between the phase noise at harmonic frequencies (the focus of previous works) and the timing jitter of photocurrent pulse edges (the focus of this

work) in the following way. The experimental study on harmonics-dependent APC coefficients[13] showed almost constant maximum APC coefficients for many harmonics and even a slight decrease at high (>10 GHz) frequency. Although the APC coefficient also strongly depends on the pulse energy at each harmonic, over many harmonics (e.g., 48 harmonics for 12-GHz bandwidth photodiode when 250-MHz comb is used), collectively on average, we can deduce that the overall APC impact will not change much or even decrease with harmonic order. When we

convert this observation into the equivalent timing change, the ATC will decrease with higher harmonics due to the smaller period at higher harmonics: in terms of timing change per amplitude change, the impact of the same APC coefficient is 40 times smaller at 10 GHz (40th harmonic) than at 250 MHz (fundamental repetition rate). This observation may qualitatively explain why the rising-edge ATC coefficient and the resulting jitter are much smaller than the falling edge. For rising edge, which is largely contributed by higher frequency harmonics, it will have smaller effective ATC coefficient by combining smaller average APC coefficient magnitude and shorter period at higher frequency. In contrast, for falling edge, which is more contributed by lower frequency harmonics, it will have much larger effective ATC coefficient by having longer period at lower frequency.

**Impact of RIN suppression on timing jitter.** In order to examine the impact of RIN and to further suppress its coupling to photocurrent pulse timing jitter, we implemented a RIN stabilization control loop in the Erbium-doped fibre amplifier (EDFA) (see Fig. 1a and Methods). Curve (ii) in Fig. 4a shows the measured RIN spectrum with RIN control by using an out-of-loop monitoring photodiode. The integrated RIN could be suppressed down to 0.0054% (r.m.s.) for 1 Hz–1 MHz offset frequency range. Note that the EDFA was operated in a non-saturated regime, which enabled effective RIN suppression by the pump current feedback. The resulting RIN followed the equivalent amplitude noise of the used voltage reference (curve (iii) in Fig. 4a) in the stabilization control loop. Curve (iv) shows the dark noise of the detection system, which shows that it is lower than the voltage reference noise (curve (iii)) and the suppressed RIN (curve (ii)). Note that the discontinuity at 100 kHz in curve (iv) is due to the higher noise floor of the radio-frequency (RF) analyser, which is used for measuring the PSD in the >100 kHz offset frequency range.

The obtained timing jitter PSDs for a 22-GHz MUTC photodiode with RIN suppression are shown in Fig. 4b. Owing to the RIN suppression, the integrated r.m.s. rising-edge jitter could be improved from 80 as (curve (iii) in Fig. 1c) to 64 as (curve (i) in Fig. 4b). While the falling-edge jitter (curve (ii)) closely follows the calculated ATC-originated jitter with suppressed RIN condition (curve (iv)), the rising-edge jitter is higher than the predicted ATC-originated jitter (curve (iii)). In the low offset frequency (<~1 kHz), it is mostly limited by the 1/$f$ flicker noise of the photodiode (line (vi)), which has a similar level with those shown in refs. [9,18,19]. In the high offset frequency (>~30 kHz), it is mostly limited by the EOS-TD measurement noise floor (curve (v)).

**Photocurrent pulse shaping for attosecond timing.** To fully exploit the ultralow timing jitter of rising edges in photocurrent pulses for attosecond electronic timing, we propose and demonstrate a photocurrent pulse shaping method. Figure 5a shows the schematic of the pulse shaping method. Since the rising-edge jitter is much lower than the falling-edge jitter, a balanced photodetector with slight timing shift ($\tau$) is used for shaping the output photocurrent pulses dominated by the rising-edge parts of two photocurrent pulses (inset of Fig. 5a). Figure 5b shows an example of measured photocurrent pulse shapes when the timing shift and relative power between positive and negative pulses are set to 50 ps and 0.375, respectively, for a 20-GHz balanced p–i–n photodetector (with ±10 V bias). Sharp photocurrent pulses can be synthesized by exploiting the steep rising-edge slopes of both photocurrent pulses from each photodiode. Figure 5c shows the measured 50%-level falling-edge jitter for different timing shifts (in the range of 40–90 ps), while the relative power level is slightly tuned at each delay condition (in the

range of 0.25–0.375) so that the shaped photocurrent pulse does not possess negative value in voltage. Note that such pulse shape without negative value is beneficial for applications such as digital clock generation by injecting current pulses to a capacitive load[35]. The falling-edge jitter is minimized at 50-ps delay and increases with larger delay since the falling-edge jitter of p–i–n photodiode increases along the temporal position toward the end of falling edge (see p–i–n photodiode case in Fig. 2b). Note that, even when the shaped pulse has negative value, we could find a similar level of minimal jitter at different delay condition. Figure 5d shows the measured 50%-level jitter PSDs at 50-ps delay and 0.375 relative power ratio condition (Fig. 5b). As expected, while keeping the ultralow rising-edge jitter (curve (ii)), falling-edge jitter is also greatly suppressed from 10 fs (curve (i)) to 270 as (curve (iii)). Note that the measured falling-edge jitter of the shaped photocurrent pulse is mostly limited by the EOS-TD detection noise floor (curve (iv); integrated jitter of 204 as) above ~30 kHz offset frequency. Thus, by photocurrent pulse shaping, attosecond timing can be obtained for both rising- and falling edges even using standard p–i–n photodiodes.

## Discussion

As the electronic system operation speed and data rate rapidly increase in recent years, lower edge-jitter clock signals are becoming more important. High-speed and high-bandwidth systems, such as analogue-to-digital converters[31], high-speed data links[32,33], clock distribution networks[37], frequency synthesizers[38] and 5G communication[34], already require tens of femtoseconds jitter and will be greatly benefited by even lower jitter in the near future. In this work, we showed a photonics-based method to generate such ultralow edge-jitter electric pulse signals. In addition to the intrinsically low jitter of mode-locked lasers and optical frequency combs well below a femtosecond[1,5–7], we could optimize the additional edge jitter in the optical-to-electronic conversion process to the sub-100-attosecond regime. In particular, we could achieve such attosecond timing using general-purpose p–i–n photodiodes, which can be also easily integrated with CMOS-compatible processes[39,40]. Although a 250-MHz mode-locked laser is used as the optical pulse source in this work, the combination of microresonator-based on-chip Kerr optical frequency combs[3,4,48,49], which have rapidly evolved in recent years, and the methods shown in this work may achieve GHz and much higher speed timing and clock generation with fully integrated electronic–photonic platforms in the near future.

## Methods

**Timing jitter measurement methods.** For the mode-locked laser (MLL), a non-linear amplifying loop mirror-based Erbium-doped fibre oscillator (FC1500-250-ULN, MenloSystems GmbH) with a repetition rate of 250 MHz is used. The comb signal is amplified by an EDFA to ~40 mW. Dispersion compensating fibre (DCF) is used in front of the EDFA to reduce the pulsewidth of optical pulses incident on the photodiode down to 200 fs. The RIN of optical pulse train can be stabilized by feedback to the pump laser diode current driver in the EDFA. The photodetected baseband signal is compared with a more stable voltage reference and the error signal is applied to the pump current driver via a PI controller. The output power of the EDFA is split by a 50:50 coupler. Half of the power is applied to the EOS-TD and the other half is applied to the photodiode-under-test through a variable optical attenuator (VOA). To change the relative temporal position between the optical pulses and the photocurrent pulses generated from a photodiode, a fibre-coupled VDL is placed in one of the optical beam paths. The VDL can scan up to 330 ps with 1 fs resolution. The electric pulse waveforms shown in Figs. 1b, 2a and 5b are measured by a 33-GHz, 128 GS/s real-time oscilloscope (Keysight, UXR0334A). Note that the effective load impedance for the measured voltage signals is 25-ohm due to the parallel connection of a 50-ohm termination to the photodiode and a 50-ohm input impedance of the oscilloscope. Three different commercial photodiodes are used for the edge-jitter test. For the MUTC photodiode, a 22-GHz MUTC photodiode (Freedom Photonics, FP1015a) and a 12-GHz MUTC photodiode (Finisar, VPDV2120-VF-FA) are used with 8-V bias voltage from an external bias-tee. A 12-GHz p–i–n photodiode (Beijing Lightsensing Technologies Ltd, LSIHPD-A12) is also used with 20-V bias voltage.

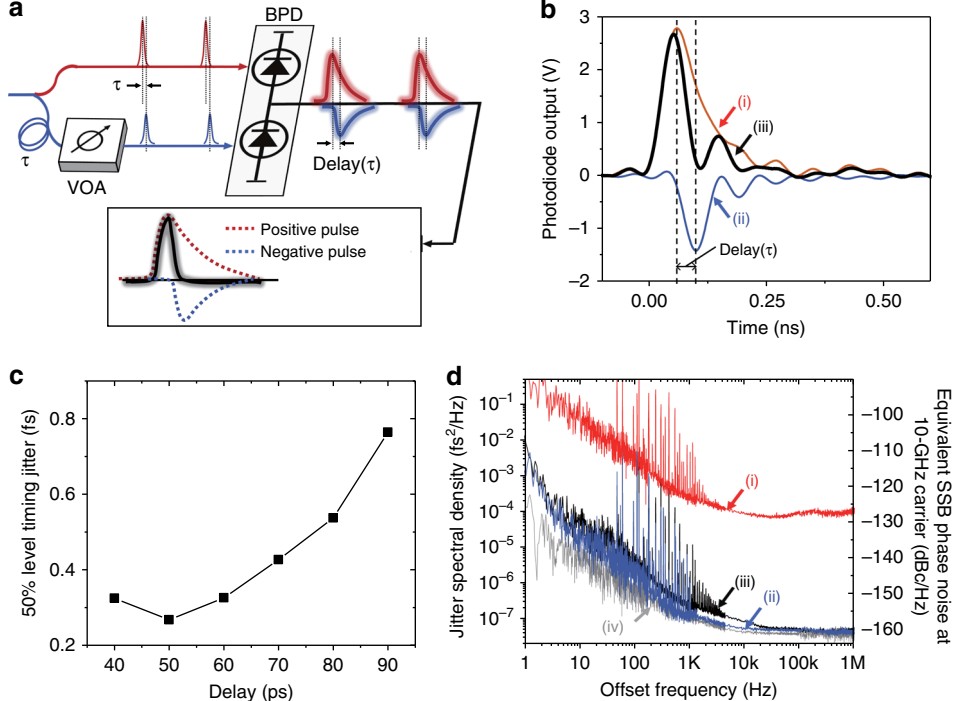

**Fig. 5 Photocurrent pulse shaping for attosecond timing. a** Concept of photocurrent pulse shaping for attosecond timing with both edges. BPD balanced photodiode, VOA variable optical attenuator. **b** Photocurrent pulse waveform (voltage across a 25-ohm impedance) of (i) positive photocurrent pulse (2 mA average current), (ii) negative photocurrent pulse (0.75 mA average current) and (iii) shaped photocurrent pulse with 50-ps delay. **c** Falling-edge timing jitter of shaped photocurrent pulses as a function of relative delay between positive and negative pulses. **d** Measured 50%-level timing jitter PSDs measured for case **b** (50-ps delay and 0.375 relative power ratio). (i) Falling-edge jitter before pulse shaping. (ii) Rising-edge jitter before pulse shaping. (iii) Falling-edge jitter after pulse shaping. (iv) EOS-TD measurement noise floor.

The relative timing jitter between optical pulses and photocurrent pulses is measured by the EOS-TD. The EOS-TD is a differential-biased fibre Sagnac-loop interferometer with an electro-optic phase modulator (PM) inside the Sagnac loop[27,28] (Fig. 1a). While electro-optic sampling has been widely used for broadband electric field detection[50], here we applied it for detecting timing fluctuations between an ultrashort optical pulse and a certain sampling point on the photocurrent pulse using a configuration similar to the fibre Sagnac-loop gyroscope[51]. The operation principle of the EOS-TD-based jitter analysis is following. When the average input optical power is $P_{in}$, the outputs from the Sagnac loop can be written as $P_1 = P_{in}\sin^2[\Delta\phi/2]$ and $P_2 = P_{in}\cos^2[\Delta\phi/2]$, where $\Delta\phi$ is the optical phase difference between the counter-propagating pulses in the Sagnac loop (Fig. 1a). The balanced photodetector output can be written as $\Delta V = -RG(P_1 - P_2) = RGP_{in}\cos[\Delta\phi]$, where $R$ (in A/W unit) is the photodiode responsivity and $-G$ (in V/A unit) is the transimpedance gain of the balanced photodetector. For differential operation of the Sagnac loop, a nonreciprocal phase bias is required. In this work, we inserted a nonreciprocal $\pi/2$ bias[52], consisting of Faraday rotators and waveplates, in the fibre loop ($\Phi_b = \pi/2$ in Fig. 1a). Then, when a unidirectional PM (e.g., a travelling-wave electro-optic LiNbO$_3$ PM) is driven by the photocurrent pulse, it can induce additional optical phase shift ($\phi_M$ in Fig. 1a). As our goal in this work is to analyse the timing fluctuation at a specific sampling point on the photocurrent pulse, we tuned the sampling point to the middle of rising edge or falling edge of photocurrent pulse using the VDL (Fig. 1a). When the sampling point is fixed, the remaining optical phase fluctuation by the photocurrent pulse is proportional to the relative timing jitter between the optical pulses and the photocurrent pulses ($\Delta t$) at the sampled point. As the timing jitter (<100 fs) is much smaller than the photocurrent length (~30 and ~250 ps for rising- and falling edges, respectively, for 12-GHz p–i–n photodiode), we can linearize the response as $\phi_M \cong K\Delta t$. The resulting phase difference between counter-propagating pulses in the Sagnac loop is then $\Delta\phi = \phi_M + \Phi_b \cong K\Delta t + \pi/2$. The balanced photodetector output is $\Delta V = RGP_{in}\cos\Delta\phi \cong RGP_{in}K\Delta t$. Note that there are sources for power imbalance between the Sagnac loop outputs in the actual setup, including the DC voltage of the sampled point on the photocurrent pulse and the circulator loss. This issue is addressed by tuning the loss between the two Sagnac-loop outputs by the variable attenuator (VOA in Fig. 1a). This will result in slight decrease in detection sensitivity compared to the ideal case, but the sensitivity was enough for characterizing tens of attoseconds jitter. The timing detection sensitivity of the EOS-TD is determined to be 1.7 mV/fs by measuring the EOS-TD voltage change when the VDL shifts relative timing up to 1 ps.

The timing jitter PSDs are measured by a fast Fourier transform (FFT) analyser (Stanford Research Systems, SR770) and an RF spectrum analyser (Agilent,

E4411B) for 1 Hz–100 kHz and 100 kHz–1 MHz offset frequency ranges, respectively. The EOS-TD measurement noise floor is determined by the product of detection sensitivity and the background voltage noise of the EOS-TD when only optical pulse train is applied (without the electric signal to the PM). The measurement floor shown in Fig. 1c is the case for the rising edge of the 22-GHz MUTC photodiode. Note that, for the falling edge, which has a slower slope compared to the rising edge, has a higher measurement floor (~1 as Hz$^{-1/2}$ = ~10$^{-6}$ fs$^2$/Hz) but far below the measured falling-edge jitter levels.

We also separately evaluated the potential impact of the ~1-m-long path length fluctuations between the two fibre paths (one to the photodiode-under-test and the other to the EOS-TD optical input after the 50:50 coupler, see Fig. 1a). For this, we measured the time-of-flight fluctuation of optical pulses transferred through a 1-km-long fibre link using an optical cross-correlator. We used a long fibre link to obtain enough timing measurement resolution. The setup is similar to the one shown in ref. [53], but instead of stabilizing the fibre link, we simply monitored the fibre length fluctuation and computed the equivalent PSD. The measured PSD is converted to the 1-m length scale, which is shown by curve (viii) in Fig. 1c. This result shows that the impact of path length fluctuation in the experiment did not limit the photodetection jitter measurements.

**Timing jitter parameter-dependence measurement methods.** To examine the relative temporal position, bias voltage, input optical power and optical pulsewidth dependence on the photocurrent pulse-edge jitters, a 12-GHz p–i–n and MUTC photodiodes are used. By changing the electro-optic sampling temporal position between optical pulses and photocurrent pulses using the VDL, relative timing jitter at any temporal position along the photocurrent pulse can be measured. Since the fall times of p–i–n and MUTC photodiodes are very different (Fig. 2a), the temporal positions along the falling edge are selected based on their amplitude, which corresponds to 10, 30, 50, 70 and 90% of the maximum amplitude of photocurrent pulse (Fig. 2b). Two different bias conditions (high and low bias voltages) for each photodiode are used to examine the impact of bias voltage. To examine the input optical power dependence (Fig. 2c), the average optical power incident on photodiodes is changed by the VOA and monitored by an inline optical power meter (for measuring average optical power) and an ammeter (for measuring average photocurrent) simultaneously. Finally, to test the impact of optical pulsewidth applied to the photodiode, the pulsewidth was changed by adjusting the length of DCF and monitored by an interferometric autocorrelator. The timing jitters at the rising- and falling edge were measured for full-width-half-maximum optical pulsewidths of 200 fs and 1, 3 and 5 ps.

**Modelling and simulation methods**. The photodiode modelling and simulation are performed to evaluate the ATC-originated jitter. Since the optical pulsewidth is much shorter than the transit time or the inverse of photodiode bandwidth, the optical pulses and the photocurrent pulses can be regarded as an input impulse and a photodiode impulse response, respectively. Hence, the transfer function can be obtained by the Fourier transform of the impulse response. As shown in Figs. 1b and 2a, the temporal pulse profiles at different average photocurrents are measured by a real-time oscilloscope (Keysight, UXR0334A). An external 50-ohm termination is used at the output of the photodiode. Therefore, the load impedance connected to the photodiode becomes 25-ohm when using the oscilloscope, whereas the EOM input impedance is 40 ohm when measuring the jitter with the EOS-TD. This impedance difference is considered and calibrated for the modelling and simulation. As a cross-check, time-delay equivalent-circuit model-based transfer functions of the photodiodes[45,54] are also constructed and compared with the measured Fourier transform of impulse responses in Fig. 3a. The ATC coefficient can be obtained through linear system simulation with a photodiode transfer function ($H(f)$).

$$\mathcal{F}\{i(t)\} = \left[\sum_k (1 + a_k)\mathcal{F}\{x(t - kT)\}H(f)\right], \quad (1)$$

where $a_k$ is the amplitude noise term at $k$th optical pulse, $\mathcal{F}$ is the Fourier transform, $T$ is the period of optical pulse train, $x(t)$ is the input signal (sech$^2$-shaped optical pulse), $i(t)$ is the output photocurrent pulse and $H(f)$ is the transfer function of the photodiode. Here, we set the amplitude change ($a_k$) as a sine wave at a certain modulation frequency with 0.034% modulation depth. By collecting the time instants of pulses within one period of sine wave that reach half of the maximum value of $i(t)$ at both rising and falling edges with 0.1-fs sampling period, the amplitude change-induced timing change can be obtained. Then, the ATC coefficient can be computed by dividing the timing change by the normalized amplitude change. We repeated this process 30 times to obtain the averaged ATC coefficient. The ATC-originated timing jitter PSD can be deduced by multiplying the ATC coefficient (obtained by the simulation outlined above) and the measured RIN spectrum, which are presented as red curves in Fig. 3d.

**Measurement of photodiode ATC coefficients**. To quantify the ATC coefficient experimentally, a separate experiment is conducted (see Fig. 3c). An AOM is placed in front of the photodiode-under-test. The timing modulation of the photodiode is monitored by the EOS-TD when the AOM is amplitude-modulated. The amplitude modulation is measured by a 1.7-MHz photodetector using a 5% tap coupler, which is located right before the photodiode. The amount of timing modulation is measured by the EOS-TD at the middle of rising- and falling edges of a photocurrent pulse. A 12-GHz p–i–n and MUTC photodiode is used at 3-mA photocurrent, with 20- and 8-V bias, respectively. The used modulation frequencies are 100, 200 and 500 Hz, 1, 2, 5, 10, 20 and 50 kHz with a modulation depth of 0.034%. As shown in Fig. 3b, we found that there is negligible frequency dependence in the measured ATC coefficients over modulation frequency, which is similar to the previous APC coefficient measurement studies[13].

**Photocurrent pulse shaping methods**. For the photocurrent pulse shaping, a 20-GHz balanced p–i–n photodetector (Optilab, BPD-20-D) is used. The optical power applied to generate positive electric pulse is 4 mW (2-mA average photocurrent), which is limited by the power handling capability of the used balanced photodiode with ±10-V bias. The relative delay and amplitude are adjusted by the VDL and the VOA, respectively, while monitoring the photocurrent pulse shape by the 33-GHz real-time oscilloscope. Then, the balanced p–i–n photodetector output is applied to the PM in the EOS-TD for the relative timing jitter measurement, compared with the optical pulses before the splitter.

## Data availability
Source data are provided with this paper.

## Code availability
The code used for making Fig. 3a, b is provided as a Source Code file. Source data are provided with this paper.

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

## Acknowledgements

This research was supported by the Samsung Research Funding and Incubation Center of Samsung Electronics under Project Number SRFC-IT1702-07. We thank Prof. Hyunyong Choi and Mr. Dohyeon Kwon for discussions.

## Author contributions

H.C. and J.K. conceived the idea and managed the project. M.H., C.A. and Y.N. performed the experiments and obtained data. M.H. performed the simulation. M.H., H.C. and J.K. analysed the data and wrote the manuscript with inputs from all authors.

## Competing interests

The authors declare no competing interests.
