## [Peer Review File · Nature Communications]

Reviewers' Comments:

Reviewer #1:

Remarks to the Author:

Review of "Attosecond electronic timing with rising edges of photocurrent pulses", by M. Hyun et al.

This manuscript reports measurements on the timing jitter of electrical pulses from high-speed photodetectors. Rising and falling edge jitter are measured (with very high precision), as opposed to the "center of mass" jitter that is more often reported in the literature. As the authors note, the distinction between the rising edge jitter and center of mass jitter is quite important for some applications. The very low rising edge jitter measured here could be useful, even with substantially higher falling edge jitter. Relative intensity noise is identified as the leading cause of the jitter, and the authors employ strong RIN suppression that greatly reduces the measured jitter. Moreover, the authors design and characterize a scheme to reduce the falling edge jitter with balanced photodetection.

The paper is well written and the measurements are thorough and quite interesting. I think the paper would of interest to the broad readership of Nature Communications. I recommend it for publication, provided the authors address the comments below.

My main request is that the authors give a bit more recognition to previous work that shows the power dependence on the photocurrent pulse shape. Examples include S. Diddams et al, Opt. Expr. vol. 17, p. 3331 (2009), W. Zhang et al, Appl. Phys. B, vol. 106 p. 301 (2012), Y. Hu et al, IEEE Photonics Journal, vol. 9, art. 5501111 (2017), and J. Davila-Rodriguez et al, Opt. Expr. vol. 26, p. 30532 (2018). All these references show and/or discuss the strong dependence on the falling edge of a photocurrent pulse with input optical power – much stronger dependence than on the rising edge. I would like the authors to consider the inclusion of one or more of these references. In light of these papers, it is not so surprising that amplitude noise on the optical pulse train has a strong impact on the falling edge.

Under short pulse illumination, MUTC detectors have a photocurrent where the bandwidth is maximized and the impulse response is minimized (J. Davila-Rodriguez et al, Opt. Expr. vol. 26, p. 30532 (2018), among others). This point is a minimum in the conversion of amplitude noise to center of mass jitter. Since most of this center of mass jitter would manifest as a falling edge jitter, I would think measurements at the minimum-width impulse response point would give lower jitter numbers on the falling edge. Did the authors look for this point? It appears that the authors only report on photocurrents above this conversion minimum point, where the noise conversion tends to be significantly higher.

Lastly, the terms amplitude to timing coefficient (ATC) and amplitude to phase coefficient (APC) are both used. Please clearly distinguish the difference between ATC and APC. This may not be immediately clear to the uninitiated.

Reviewer #2:

Remarks to the Author:

In their manuscript entitled "Attosecond electronic timing with rising edges of photocurrent pulses", Hyun et al. report on a study where a train of picosecond electrical pulses is generated by coupling a modelocked Er: fiber laser into various broadband photodetectors. Exploiting the electro-optic effect, the authors characterize the timing jitter between the optical and electrical pulse trains and find

remarkably low values down to the range below 100 as. The information that the timing jitter is especially low in the rising edge of the electrical pulse enables generation of broadband periodic electrical waveforms which are important for various tasks in contemporary high-speed electronics, data processing and information technology. The authors explore the best conditions for low jitter in terms of photodiode biasing and optical pulse energy. They demonstrate that the timing jitter predominantly arises from amplitude fluctuations of the optical pulse train. This finding is exploited to further enhance the quality by active noise control in the Er:amplifier of the optical system. Also, the authors demonstrate that elimination of the trailing edge of the optical pulse by generation of the difference between two wave forms with adjustable amplitude and time delay almost readily suppresses the timing jitter in the trailing edge. The paper is scholarly written and reports interesting results which are relevant for front-end contemporary optoelectronics and its applications. Therefore, this referee suggests publication of the study in Nature Communications. Beforehand, the authors should further enhance their manuscript by working on the following list of questions and comments:

I) One of the main points motivating the present contribution is the fact that most previous work has focussed on characterizing the stability of a single frequency out of the harmonics spectrum of the electrical pulse train. Naively, from the results of the present paper, one would expect changes in the phase noise of each harmonic component depends on how much contributions are coming from the rising (mostly high-frequency components) or falling (fundamental and other low-order harmonics?). The authors might consider to briefly mention this context together with some analysis that should be easy for them to access.

II) The authors convincingly show that the amplitude noise on their optical pulse train represents the main source for phase noise in the electrical output. From this, it seems straightforward to at least qualitatively understand that the rising edge contains less jitter than the falling one: Basically, the ratio is given just by the ratio between the temporal slopes of the amplitude. Is this simple picture really correct or is there more physics behind it? Please comment briefly in the paper.

III) The brief sketch in the Methods section and Refs. 21 + 22 are not really adequate to explain the principle of the characterization method to a general audience. For an example of a tutorial introduction to the principle of ultrafast electro-optic sampling, please see Eur. J. Phys. 38, 024003 (2017).

IV) The variations in electronic pulse shape the authors observe upon changes of the biasing voltage on the photodetectors and number of photons absorbed per pulse are demonstrated to directly result in the behaviour of the timing jitter in the electrical pulse train. This entire context is simulated based on a model exploiting effective models for each device. Nevertheless, the physical origin for those effects is fairly clear when considering high-field transport in III-V semiconductors in combination with a finite screening of the biasing electric field due to the separation of photogenerated electron-hole pairs. Specifically, there is a sharp transition from ballistic to diffusive transport as soon as electrons are gaining energy beyond the intervalley threshold which is fairly advantageous in InGaAs. Please have a look at Phys. Rev. Lett. 82, 5140 (1999) and Phys. Rev. B61, 16642 (2000). In case the width of the high-field zone of their photodetectors is known, the authors would be able to qualitatively discuss the resulting dependences. This would further enhance the value of the paper for a general audience.

V) Interestingly, the data for the MUTC diode under 16 V of reverse bias are not included in Fig. 2(c). Nevertheless, a consistency check with the lower degree of change of noise expected from the pulse shape in Fig. 2(a) might be worth to mention.

VI) At the bottom of p. 6 of the manuscript, the authors discuss changes in the noise performance

when varying the optical pulse duration between 150 fs and 1 ps. In a linear regime, those effects are surprising given the fact that both pulse durations are far below the temporal response of the electronic system. Have the authors considered nonlinear processes such as e.g. two-photon absorption?

VII) In the lower half of p. 9, the authors state that "the shaped photocurrent pulse does not possess negative value in voltage". Why is it important to avoid negative voltages in the pulse form?

VIII) The second sentence of the discussion on p. 10 is almost identical to the abstract. Please avoid unnecessary repetitions to save space for further enhancement of the scientific content.

IX) In the first section of the Methods section, the authors state that "The RIN of optical pulse train can be stabilized by feedback to the pump laser diode current driver in the EDFA.". Actually, this fact is surprising because usually, such femtosecond Er: fiber amplifiers are pumped close to saturation, thus eliminating amplitude noise and rendering active stabilization with the pump power a highly nonlinear endeavour. Can the authors comment on the specific operational setting of their laser system?

X) In Fig. 4(a), please include the dark noise of the detection system in order to give the reader an impression in how far the stabilized result is limited by this contribution.

Reviewer #3:

Remarks to the Author:

Attosecond electronic timing with rising edges of photocurrent pulses, submitted to Nature Communications by Minji Hyun et al.

This work describes how to generate precise timing signals from a femtosecond pulse train. This is an important technology because the clock output of a modern optical atomic clock is of this type. Moreover, extremely stable radio frequency signals are obtained with this technology. This work sheds more light on the limitations of the fidelity to extract these stable timing signals and describes how to surpass these limitations. Therefore, I believe it is suitable for being published in Nature Communications. The manuscript is very well written. I have only a few comments that might help to improve it:

- How is the noise floor measured, that is presented in fig. 1c? By turning off the laser? By blocking either the optical or electrical path? In both cases the path length jitters is not included in the noise floor. 2nm path length fluctuations correspond to 60as.

- Amplitude to timing jitter is claimed to be the dominant noise contribution. However, it seems like this conclusion is obtained only from modelling together with the measured pulse response of the photodiode. Instead of plotting the distribution of the "amplitude noise" and (computed) "timing error" separately in fig.3a one might plot the measured "amplitude noise" vs. the measured "timing error" to verify the correlations.

- I suppose the measured RIN reduction presented in fig.4a is measured on loop. As in-loop measurements usually don't allow to draw conclusions on the measured system, this should be clearly stated.

- Page 4: a 40dB reduction of the PSD means a 20dB reduction of the timing jitter. Maybe this should be said explicitly to not mislead the reader.

- fig1a: inspecting the arrows at the Sagnac interferometer I would conclude that there are only exiting arrows. One may wonder how it is possible that the Sagnac has only light leaving it but no inputs.

- Fig.1b: it is impossible to see the 60as timing error on a ns time axis. The resolution obviously exists as the spectral domain data in fig.1c shows. It might be helpful to see a magnified time domain picture that shows the 60as timing jitter.

- Fig2b is a mystery to me. The caption says the pulse position is plotted against the dependence on timing, but the axis read timing jitter vs. normalized amplitude. Why is the timing jitter of the order of 20fs instead of the claimed 60as? What is the meaning of the dashed box at the lower left corner and the arrow? I have no clue what this all means and strongly recommend to rework this graph.

- abstract: "... using low-cost and silicon-photonics-integrable ...". I don't think low cost is a scientific argument. This is even more so when we speak about the cost of a photodetector as compared to the cost of a frequency comb.

- Equation (1) on page 14: It might help to define the function F , i.e, say that it is the Fourier Transform.

Point-by-point response to Reviewer #1's comments

We thank Reviewer #1 for constructive suggestions and comments on our manuscript.

This manuscript reports measurements on the timing jitter of electrical pulses from high-speed photodetectors. Rising and falling edge jitter are measured (with very high precision), as opposed to the “center of mass” jitter that is more often reported in the literature. As the authors note, the distinction between the rising edge jitter and center of mass jitter is quite important for some applications. The very low rising edge jitter measured here could be useful, even with substantially higher falling edge jitter. Relative intensity noise is identified as the leading cause of the jitter, and the authors employ strong RIN suppression that greatly reduces the measured jitter. Moreover, the authors design and characterize a scheme to reduce the falling edge jitter with balanced photodetection.

The paper is well written and the measurements are thorough and quite interesting. I think the paper would of interest to the broad readership of Nature Communications. I recommend it for publication, provided the authors address the comments below.

Point 1-1

My main request is that the authors give a bit more recognition to previous work that shows the power dependence on the photocurrent pulse shape. Examples include S. Diddams et al, Opt. Expr. vol. 17, p. 3331 (2009), W. Zhang et al, Appl. Phys. B, vol. 106 p. 301 (2012), Y. Hu et al, IEEE Photonics Journal, vol. 9, art. 5501111 (2017), and J. Davila-Rodriguez et al, Opt. Expr. vol. 26, p. 30532 (2018). All these references show and/or discuss the strong dependence on the falling edge of a photocurrent pulse with input optical power – much stronger dependence than on the rising edge. I would like the authors to consider the inclusion of one or more of these references. In light of these papers, it is not so surprising that amplitude noise on the optical pulse train has a strong impact on the falling edge.

We added the suggested papers in the reference list of the revised manuscript (refs. 13, 21-23, underlined), and also added the following discussion in page 3 (lines 51-55): “Note that, as some relevant researches on photocurrent pulse properties, previous studies on the input optical power-dependent photocurrent pulse shape showed much stronger dependence on the falling edge than the rising edge^{13,21-23}, which suggests different behaviour in excess noise between rising- and falling-edges of photocurrent pulses.”

Point 1-2

Under short pulse illumination, MUTC detectors have a photocurrent where the bandwidth is maximized and the impulse response is minimized (J. Davila-Rodriguez et al, Opt. Expr. vol. 26, p. 30532 (2018), among others). This point is a minimum in the conversion of amplitude noise to center of mass jitter. Since most of this center of mass jitter would manifest as a falling edge jitter, I would think measurements at the minimum-width impulse response point would give lower jitter numbers on the falling edge. Did the authors look for this point? It appears that the

authors only report on photocurrents above this conversion minimum point, where the noise conversion tends to be significantly higher.

Following the reviewer's suggestion, we applied lower input optical power (pulse energy) to find the minimum-width impulse response point. However, we could not find such point or sharp reduction in jitter in our case. The main reason is that, at such low photocurrent level (e.g., below 1 mA), the EOS-TD did not have enough timing resolution and showed almost same noise floors for different power levels (we tested 0.25 mA, 0.4 mA, 0.55 mA, 0.7 mA, 0.85 mA). More specifically, when the photocurrent was less than ~ 0.5 mA, the balanced photodetector in the EOS-TD was limited by the shot noise floor, and thus could not distinguish the difference in jitter PSDs at all. We believe that the improvement of timing resolution and more careful study in such low current regime will be an interesting future research topic. We added the following discussion in page 7 (lines 145-152): "Note that we also tried to measure the falling-edge jitter at lower photocurrent levels (e.g., <1 mA) to find the recently studied minimum-width impulse response point²³, where the impulse response and the coupling of amplitude noise to centre-of-mass jitter are minimized. However, due to the limited EOS-TD resolution in lower current regime, the measurements were limited by the EOS-TD resolution itself and could not distinguish differences in jitter PSDs. Improvement of EOS-TD resolution and more careful jitter study in such low photocurrent regime will be an interesting future research topic."

Point 1-3

Lastly, the terms amplitude to timing coefficient (ATC) and amplitude to phase coefficient (APC) are both used. Please clearly distinguish the difference between ATC and APC. This may not be immediately clear to the uninitiated.

We added the following explanation in page 6 of the revised manuscript (lines 120-125): "Note that, in the following discussions, we will use the ATC (i.e., timing change per amplitude change) instead of the APC (i.e., phase change per amplitude change) which has been more widely used for the single-tone microwave extraction research^{17,41}. Since the photocurrent pulse itself contains many frequency components, the impact of amplitude noise can be quantified more meaningfully by considering the timing change rather than the phase change."

Point-by-point response to Reviewer #2's comments

We thank Reviewer #2 for constructive suggestions and comments on our manuscript.

In their manuscript entitled "Attosecond electronic timing with rising edges of photocurrent pulses", Hyun et al. report on a study where a train of picosecond electrical pulses is generated by coupling a modelocked Er: fiber laser into various broadband photodetectors. Exploiting the electro-optic effect, the authors characterize the timing jitter between the optical and electrical pulse trains and find remarkably low values down to the range below 100 as. The information that the timing jitter is especially low in the rising edge of the electrical pulse enables generation of broadband periodic electrical waveforms which are important for various tasks in contemporary high-speed electronics, data processing and information technology. The authors explore the best conditions for low jitter in terms of photodiode biasing and optical pulse energy. They demonstrate that the timing jitter predominantly arises from amplitude fluctuations of the optical pulse train. This finding is exploited to further enhance the quality by active noise control in the Er: amplifier of the optical system. Also, the authors demonstrate that elimination of the trailing edge of the optical pulse by generation of the difference between two wave forms with adjustable amplitude and time delay almost readily suppresses the timing jitter in the trailing edge. The paper is scholarly written and reports interesting results which are relevant for front-end contemporary optoelectronics and its applications. Therefore, this referee suggests publication of the study in Nature Communications. Beforehand, the authors should further enhance their manuscript by working on the following list of questions and comments:

Point 2-1

One of the main points motivating the present contribution is the fact that most previous work has focussed on characterizing the stability of a single frequency out of the harmonics spectrum of the electrical pulse train. Naively, from the results of the present paper, one would expect changes in the phase noise of each harmonic component depends on how much contributions are coming from the rising (mostly high-frequency components) or falling (fundamental and other low-order harmonics?). The authors might consider to briefly mention this context together with some analysis that should be easy for them to access.

The main findings of our work are: (a) timing jitter at the rising- and falling-edges of photocurrent pulses is mostly limited by the amplitude-to-timing conversion (ATC) and (b) the falling-edge ATC coefficient is much larger than the rising-edge ATC coefficient, which results in much larger falling-edge jitter than the rising-edge jitter.

From this finding, we can make a link between the phase noise at each harmonic frequency (the focus of previous works) and the timing jitter of photocurrent pulse edges (the focus of this work) in the following way. As the reviewer rightly pointed out, the rising-edge is mostly contributed by the higher harmonics while the falling-edge is mostly contributed by the lower harmonics. Although there are only few studies that experimentally characterized the harmonics-dependent amplitude-to-phase conversion (APC) coefficients, the study by SYRTE [Zhang et al, Appl.

Phys. B 106, 301 (2012), which is ref. 13] showed that the overall APC coefficient magnitudes are almost constant (max <1 rad) for many harmonics. It also showed a decrease in APC coefficient magnitude to <0.2 rad at very high (11.55 GHz) frequency. Although the APC coefficient also strongly depends on the optical power and pulse energy at each harmonic, over many harmonics (e.g., 48 harmonics for 12-GHz bandwidth photodiode when 250-MHz comb is used), collectively on average, we can deduce that the overall APC impact will not change much or even decreased with harmonic order. When we convert this observation into the equivalent timing change, the amplitude-to-timing conversion will decrease with higher harmonics due to the smaller period at higher frequency (i.e., ATC coefficient [s/(dP/P) unit] = APC coefficient [rad/(dP/P)] \times T_n/2 π [s/rad], where T_n is the period of the n-th harmonic, T_n = 1/nf_{rep}). Thus, in terms of timing change per amplitude change, the impact of the same APC coefficient is 40 times smaller at 10-GHz (40th harmonic) than at 250-MHz (fundamental repetition-rate).

This observation may qualitatively explain why the rising-edge ATC and the resulting timing jitter are much smaller than the falling-edge ATC and timing jitter. For rising-edge, which is largely contributed by higher frequency harmonics (e.g., >9 GHz), will have smaller ATC by combining smaller average APC magnitude and smaller period at higher frequency. In contrast, for falling-edge, which is contributed by lower frequency harmonics, will have much larger ATC by having longer period at lower frequency.

We added the above discussion on the link between harmonic phase noise and pulse-edge timing jitter in the “Impact of amplitude-to-timing conversion on timing jitter” section in page 10 (lines 200-217): “From this finding, we can also connect a link between the phase noise at harmonic frequencies (the focus of previous works) and the timing jitter of photocurrent pulse edges (the focus of this work) in the following way. The experimental study on harmonics-dependent APC coefficients¹³ showed almost constant maximum APC coefficients for many harmonics and even a slight decrease at high (>10 GHz) frequency. Although the APC coefficient also strongly depends on the pulse energy at each harmonic, over many harmonics (e.g., 48 harmonics for 12-GHz bandwidth photodiode when 250-MHz comb is used), collectively on average, we can deduce that the overall APC impact will not change much or even decrease with harmonic order. When we convert this observation into the equivalent timing change, the amplitude-to-timing conversion will decrease with higher harmonics due to the smaller period at higher harmonics: in terms of timing change per amplitude change, the impact of the same APC coefficient is 40 times smaller at 10-GHz (40th harmonic) than at 250-MHz (fundamental repetition-rate). This observation may qualitatively explain why the rising-edge ATC coefficient and the resulting jitter are much smaller than the falling-edge. For rising-edge, which is largely contributed by higher frequency harmonics, it will have smaller effective ATC coefficient by combining smaller average APC coefficient magnitude and shorter period at higher frequency. In contrast, for falling-edge, which is more contributed by lower frequency harmonics, it will have much larger effective ATC coefficient by having longer period at lower frequency.”

Point 2-2

The authors convincingly show that the amplitude noise on their optical pulse train represents the main source for phase noise in the electrical output. From this, it seems straightforward to at least qualitatively understand that the rising edge contains less jitter than the falling one: Basically, the ratio is given just by the ratio between the temporal slopes of the amplitude. Is this simple picture really correct or is there more physics behind it? Please comment briefly in the paper.

We believe that our response and revision for Point 2-1 mostly answer this question as well. It is indeed true that the steeper slope of rising-edge is responsible for much smaller jitter, but it did not scale by the simple slope ratio between rising- and falling-edges. The measured jitter ratio was much larger (~70 times) than the slope ratio, and as outlined in the response for Point 2-1, it seems that much smaller amplitude-to-timing conversion (ATC) for rising-edge is the reason. The steep slope of rising-edge is contributed mostly by the high harmonic frequency components (e.g., >9 GHz), and the ATC magnitude in such high frequency range is much smaller than the lower frequency range. Our measured ATC coefficient of rising-edge was indeed ~50 times smaller than that of falling-edge, and we could well explain our measured jitter spectra by multiplying the ATC coefficients with measured RIN spectra (as shown in Fig. 3d).

Point 2-3

The brief sketch in the Methods section and Refs. 21 + 22 are not really adequate to explain the principle of the characterization method to a general audience. For an example of a tutorial introduction to the principle of ultrafast electro-optic sampling, please see Eur. J. Phys. 38, 024003 (2017).

We added more comprehensive description of the principle of our characterization method (EOS-TD) in Methods section (lines 312-340 in pages 15-16): “While electro-optic sampling has been widely used for broadband electric field detection⁵⁰, here we applied it for detecting timing fluctuations between an ultrashort optical pulse and a certain sampling point on the photocurrent pulse using a configuration similar to the fibre Sagnac-loop gyroscope⁵¹. The operation principle of the EOS-TD-based jitter analysis is following. When the average input optical power is P_{in} , the outputs from the Sagnac-loop can be written as $P_1 = P_{in}\sin^2[\Delta\phi/2]$ and $P_2 = P_{in}\cos^2[\Delta\phi/2]$, where $\Delta\phi$ is the optical phase difference between the counter-propagating pulses in the Sagnac-loop (Fig. 1a). The balanced photodetector output can be written as $\Delta V = -RG(P_1 - P_2) = RGP_{in}\cos[\Delta\phi]$, where R (in A/W unit) is the photodiode responsivity and -G (in V/A unit) is the transimpedance gain of the balanced photodetector. For differential operation of the Sagnac-loop, a nonreciprocal phase bias is required. In this work, we inserted a nonreciprocal $\pi/2$ bias⁵², consisting of Faraday rotators and waveplates, in the fibre loop ($\Phi_b = \pi/2$ in Fig. 1a). Then, when a unidirectional PM (e.g., a travelling-wave electro-optic LiNbO₃ PM) is driven by the photocurrent pulse, it can induce additional optical phase shift (ϕ_M in Fig. 1a). As our goal in this work is to analyse the timing fluctuation at a specific sampling point on the photocurrent pulse,

we tuned the sampling point to the middle of rising-edge or falling-edge of photocurrent pulse using the variable optical delay-line (VDL in Fig. 1a). When the sampling point is fixed, the remaining optical phase fluctuation by the photocurrent pulse is proportional to the relative timing jitter between the optical pulses and the photocurrent pulses (Δt) at the sampled point. As the timing jitter (less than 100 fs) is much smaller than the photocurrent length (~ 30 ps and ~ 250 ps for rising- and falling-edges, respectively, for 12-GHz p-i-n photodiode), we can linearize the response as $\phi_M \cong K\Delta t$. The resulting phase difference between counter-propagating pulses in the Sagnac-loop is then $\Delta\phi = \phi_M + \Phi_b \cong K\Delta t + \pi/2$. The balanced photodetector output is $\Delta V = RGP_{in}\cos\Delta\phi \cong RGP_{in}K\Delta t$. Note that there are sources for power imbalance between the Sagnac-loop outputs in the actual setup, including the DC voltage of the sampled point on the photocurrent pulse and the circulator loss. This issue is addressed by tuning the loss between the two Sagnac-loop outputs by the variable attenuator (VOA in Fig. 1a). This will result in slight decrease in detection sensitivity compared to the ideal case, but the sensitivity was enough for characterizing tens of attoseconds jitter.” We also updated Figure 1a for better explaining the EOS-TD operation.

Point 2-4

The variations in electronic pulse shape the authors observe upon changes of the biasing voltage on the photodetectors and number of photons absorbed per pulse are demonstrated to directly result in the behaviour of the timing jitter in the electrical pulse train. This entire context is simulated based on a model exploiting effective models for each device. Nevertheless, the physical origin for those effects is fairly clear when considering high-field transport in III-V semiconductors in combination with a finite screening of the biasing electric field due to the separation of photogenerated electron-hole pairs. Specifically, there is a sharp transition from ballistic to diffusive transport as soon as electrons are gaining energy beyond the intervalley threshold which is fairly advantageous in InGaAs. Please have a look at Phys. Rev. Lett. 82, 5140 (1999) and Phys. Rev. B61, 16642 (2000). In case the width of the high-field zone of their photodetectors is known, the authors would be able to qualitatively discuss the resulting dependences. This would further enhance the value of the paper for a general audience.

We added the discussion on previous studies on ultrafast charge transports in high-speed photodiodes in the introduction (page 3, lines 55-57): “There were also previous studies on elucidating ultrafast photocharge transports in high-speed photodiodes using electro-optic sampling techniques²⁴⁻²⁶, showing strong dependence on the bias voltages (electric field) and optical pulse energy.”

Regarding making the link between the ultrafast charge transport phenomena and this work, we find it a bit difficult to make a direct connection. It is so because the previous studies were on studying electron transport dynamics happening within 1 ps time scale, whereas our studies were on photocurrent pulses stretching over several hundreds ps time scale, where the photocurrent width is mostly limited by the RC time constant of photodiode with ~ 10 -20 GHz bandwidth. More specifically, faster rising-edge did not have much dependence on bias voltage

(electric field) or input optical power, whereas slower falling-edge had a strong dependence on bias voltage and input optical power. We can make a qualitative explanation on falling-edge jitter dependence based on the fact that the overall photocurrent pulse length, which is mostly limited by the shape and length of falling-edge, has a strong dependence on bias voltage and input optical power. We believe that these effects can be best explained by the space-charge-screening effect. Higher bias voltage leads to shorter falling-edge length and lower jitter by relieving the space-charge-screening effect. On the other hand, higher input optical power and pulse energy leads to stronger space-charge-screening effect, which induces slower response and more distorted falling-edge, eventually leading to higher jitter. We added more explanations in pages 6-7 (lines 133-134, 144): “the jitter is reduced by increasing the bias voltage since higher bias voltage relieves the space-charge-screening effects in a photodiode... the falling-edge jitter tends to increase with optical power because the nonlinearity of a photodiode is grown by stronger space-charge-screening effects.”

Point 2-5

Interestingly, the data for the MUTC diode under 16 V of reverse bias are not included in Fig. 2(c). Nevertheless, a consistency check with the lower degree of change of noise expected from the pulse shape in Fig. 2(a) might be worth to mention.

Following the reviewer’s suggestion, we took additional measurements and added the data for 16 V bias of MUTC photodiode in Fig. 2c. As expected, higher bias could reduce the falling-edge jitter by a factor of ~2. To make the figure more complete, we included all four cases (low bias and high bias for pin photodiode; low bias and high bias for MUTC photodiode) in the revised Fig. 2c.

Point 2-6

At the bottom of p. 6 of the manuscript, the authors discuss changes in the noise performance when varying the optical pulse duration between 150 fs and 1 ps. In a linear regime, those effects are surprising given the fact that both pulse durations are far below the temporal response of the electronic system. Have the authors considered nonlinear processes such as e.g. two-photon absorption?

In fact, there were previous studies that observed a similar degradation in phase noise when the optical pulsewidth becomes longer. For example, a study by SYRTE [Xie, *EFTF/IFCS 2017*; new ref. 43] showed that longer optical input pulse led to higher phase noise level, where the noise floor is degraded by 5 dB when the pulsewidth increases from 0.8 ps to 2.6 ps. There was also a computational study by NIST [Hu, *IEEE PJ 9, 1 (2017)*; new ref. 22] that showed that the impulse response of photodiode changes even when the optical pulsewidth is much shorter than the photodiode response time.

What we observed in our study is that, when the optical pulsewidth becomes significantly longer (e.g., few ps), the rising-edge timing jitter can be degraded. Note that all the high-performance results shown in the paper were obtained when the pulse width is minimized to ~200 fs.

Compared to the original manuscript, in this revision, we extended our measurements for different optical pulsewidths from 200 fs to 5 ps and added a new Fig. 2d to show the timing jitter PSD results. For falling-edge, the jitter did not change even when the pulsewidth becomes longer. In contrast, the rising-edge jitter degraded significantly, ~7 dB and ~3 dB for 12-GHz MUTC and p-i-n photodiodes, respectively, when the optical pulsewidth is increased from 200 fs to 5 ps.

We attribute the optical pulsewidth dependence to a result of dynamic changes in the space-charge-screening effect, similar to the previous study on the excess noise from photocarrier scattering [Sun et al, PRL 113, 203901 (2014); new ref. 44]. If the optical pulsewidth is narrower, the photocarriers are likely to experience the same space-charge-screening effect, which results in a smaller variation in transit time and lower timing jitter at the rising edge.

We added this following explanation in pages 7-8 (lines 152-160): “Finally, we also examined the timing jitter dependence on the input optical pulsewidth in the range of 200 fs to 5 ps. For both types of photodiodes, the timing jitter of falling edge is almost independent on the input pulsewidth. For rising edge jitter, some degradation with long optical pulsewidth is observed (Fig. 2d) similar to previous result of microwave phase noise⁴³. This optical pulsewidth dependence might be a result of dynamic changes in the space-charge-screening effect, which is similar to the previous study on the excess noise from photocarrier scattering⁴⁴. When the optical pulsewidth is shorter, the photocarriers are likely to experience the same space-charge-screening effect, which results in a smaller variation in transit time and lower timing jitter at the rising edge.”

Point 2-7

In the lower half of p. 9, the authors state that "the shaped photocurrent pulse does not possess negative value in voltage". Why is it important to avoid negative voltages in the pulse form?

In principle, there is no reason to avoid negative voltage. Even when the shaped pulse has negative value, one can find a proper delay condition where the falling-edge jitter is minimized to the similar level (see below for comparing the two cases, with and without negative voltage in the shaped pulse, which shows similar ~270-as minimal jitter).

Without negative voltage in shaped pulse

With negative voltage in shaped pulse

One application (what we had in mind), which can be benefited by avoiding negative voltage, is the digital clock waveform generation. As shown in Debaes, IEEE JSTQE 9, 400 (2003) [ref 35], by injecting the photocurrent pulses with opposite polarity to a capacitive load, one can integrate the pulses and make rectangular-shaped digital clock signals. When making such digital clocks, negative value in the pulse can distort the integrated clock edge shape. This is why we intended to make a synthesized pulse without negative voltage. We added this explanation in page 12 (lines 254-255, 258-259): “Note that such pulse shape without negative value is beneficial for applications such as digital clock generation by injecting current pulses to a capacitive load³⁵... Note that, even when the shaped pulse has negative value, we could find a similar level of minimal jitter at different delay condition.”

Point 2-8

The second sentence of the discussion on p. 10 is almost identical to the abstract. Please avoid unnecessary repetitions to save space for further enhancement of the scientific content.

We removed the listing of potential applications in the abstract to avoid unnecessary repetition.

Point 2-9

In the first section of the Methods section, the authors state that "The RIN of optical pulse train can be stabilized by feedback to the pump laser diode current driver in the EDFA.". Actually, this fact is surprising because usually, such femtosecond Er: fiber amplifiers are pumped close to saturation, thus eliminating amplitude noise and rendering active stabilization with the pump power a highly nonlinear endeavour. Can the authors comment on the specific operational setting of their laser system?

In fact, we operated the EDFA in a non-saturated condition, and the RIN suppression by feedback control worked fairly well. Note that, when we raise the pump power and make the EDFA saturated, as the reviewer mentioned, the RIN suppression indeed did not work as

effectively. The resulting RIN was limited by the amplitude noise of voltage reference in the feedback loop. We added the equivalent RIN of the used voltage reference as curve (iii) in Fig. 4a. We added the following explanation in page 11 (lines 224-227): “Note that the EDFA was operated in a non-saturated regime, which enabled effective RIN suppression by the pump current feedback. The resulting RIN followed the equivalent amplitude noise of the used voltage reference (curve (iii) in Fig. 4a) in the stabilization control loop.”

Point 2-10

In Fig. 4(a), please include the dark noise of the detection system in order to give the reader an impression in how far the stabilized result is limited by this contribution.

In the revised Fig. 4a, we added the dark noise of the detection system (curve (iv)). It is much lower than the equivalent noise of the voltage reference (curve (iii)) and the stabilized RIN (curve (ii)). Note that the discontinuity at 100 kHz for dark noise floor (curve (iv)) is caused by higher background noise level of the used RF analyzer. As also written in the Methods, we used an FFT analyzer and RF analyzer for measuring PSDs in the <100 kHz and >100 kHz offset frequency, respectively. While the FFT analyzer noise floor was lower than the photodetector noise floor, the RF analyzer noise floor was higher and it limited the dark noise floor for >100 kHz range. We added the following explanation in page 11 (lines 227-230): “Curve (iv) shows the dark noise of the detection system, which shows that it is lower than the voltage reference noise (curve (iii)) and the suppressed RIN (curve (ii)). Note that the discontinuity at 100 kHz in curve (iv) is due to the higher noise floor of the RF analyser, which is used for measuring the PSD in the >100 kHz offset frequency range.”

Point-by-point response to Reviewer #3's comments

We thank Reviewer #3 for constructive suggestions and comments on our manuscript.

This work describes how to generate precise timing signals from a femtosecond pulse train. This is an important technology because the clock output of a modern optical atomic clock is of this type. Moreover, extremely stable radio frequency signals are obtained with this technology. This work sheds more light on the limitations of the fidelity to extract these stable timing signals and describes how to surpass these limitations. Therefore, I believe it is suitable for being published in Nature Communications. The manuscript is very well written. I have only a few comments that might help to improve it:

Point 3-1

How is the noise floor measured, that is presented in fig. 1c? By turning off the laser? By blocking either the optical or electrical path? In both cases the path length jitters is no included in the noise floor. 2nm path length fluctuations correspond to 60as.

The noise floor is measured when the optical input to the photodiode is blocked and no electrical signal is applied to the EOS-TD. Actually, this was already written in Methods section of the original manuscript (Lines 345-348 of page 16). To make it easier to follow in the main manuscript, we also put this explanation in the Results section as well (in p. 5, lines 103-104): “Note that the EOS-TD measurement noise floor (curve (vii)) is determined when the optical input to the photodiode-under-test is blocked and no electrical signal is applied to the EOS-TD.”

Regarding the path length jitter issue, as the reviewer pointed out, it is not included in the noise floor (curve (vii) in Fig. 1c). Indeed, there was ~ 1 m fiber path length mismatch between optical and electrical paths in the experiment setup, which can impact the jitter measurement result. Thus, to evaluate the impact of path length mismatch jitter, we measured the fluctuation in pulse time-of-flight (TOF) of a ~ 1 -km-long fiber link using an optical cross-correlator. Note that we used a long fiber link for estimating the fiber jitter to obtain enough measurement resolution (i.e., short fiber delay does not have enough amount of timing fluctuation that can be properly resolved). When we convert the measured fiber jitter PSD to the path length mismatch in the experiment (~ 1 m), it corresponds to the curve (viii) in the figure below (revised Fig. 1c).

The potential impact of path length jitter is much lower than the measured rising-edge jitter results (curves (i)-(iii)), therefore we can conclude that the photodiode jitter measurement results are not limited by the path length mismatch jitter. In the revised manuscript, we (1) added curve (viii) in Fig. 1c to show the estimated path length jitter PSD, (2) added discussion on this issue in page 5 (lines 105-108): “We also separately estimated

the impact of path length mismatch between the two fibre paths (i.e., one path to the photodiode-under-test, the other path to the optical input of the EOS-TD in Fig. 1a; see Methods), and as shown by curve (viii) in Fig. 1c, it is much lower than the rising-edge jitter measurement results.”, and (3) added the method for path length jitter estimation experiment in Methods (page 17, lines 352-361): “We also separately evaluated the potential impact of the ~1-m-long path length fluctuations between the two fibre paths (one to the photodiode-under-test and the other to the EOS-TD optical input after the 50:50 coupler, see Fig. 1a). For this, we measured the time-of-flight fluctuation of optical pulses transferred through a 1-km-long fibre link using an optical cross-correlator. We used a long fibre link to obtain enough timing measurement resolution. The setup is similar to the one shown in ref. ⁵³, but instead of stabilizing the fibre link, we simply monitored the fibre length fluctuation and computed the equivalent PSD. The measured PSD is converted to the 1-m length scale, which is shown by curve (viii) in Fig. 1c. This result shows that the impact of path length fluctuation in the experiment did not limit the photodetection jitter measurements.”

Point 3-2

Amplitude to timing jitter is claimed to be the dominant noise contribution. However, it seems like this conclusion is obtained only from modelling together with the measured pulse response of the photodiode. Instead of plotting the distribution of the “amplitude noise” and (computed) “timing error” separately in fig.3a one might plot the measured “amplitude noise” vs. the measured “timing error” to verify the correlations.

Following the reviewer suggestion, we also obtained the amplitude-to-timing conversion (ATC) coefficient by measurements using a similar method for obtaining the amplitude-to-phase conversion (APC) coefficients in the previous studies (e.g., Rouvalis, JLT 32, 3810 (2014); new ref. 47). The ATC measurement method is shown in the revised Fig. 3c. The amplitude modulation was applied by the acousto-optic modulator (AOM) and the amplitude and timing changes were measured by the photodetector and the EOS-TD, respectively. We applied 100 Hz, 200 Hz, 500 Hz, 1 kHz, 2 kHz, 5 kHz, 10 kHz, 20 kHz and 50 kHz modulation with 0.034% modulation depth, and measured the ratio between amplitude change and timing change, which gives the ATC coefficient. The measured (and previously simulated) ATC coefficient results are plotted in the revised Fig. 3b. The simulated and measured coefficients have a fairly good agreement each other. We also updated Fig. 3d (comparison of the RIN-originated jitter PSDs and actual jitter PSDs) to show both the predicted jitter PSDs using simulated and measured ATC coefficients.

To include these changes, we changed the previous “Modelling and simulation of photocurrent pulse jitter” subsection to “Impact of amplitude-to-timing conversion on timing jitter” to include both ATC simulation and measurement results. We added the following discussion in p. 9 (lines 185-192): “Then, the ATC coefficients were experimentally measured by a method similar to the APC coefficients measurement methods⁴⁷ (Fig. 3c; see Methods). The acousto-optic modulator (AOM) is used for amplitude modulation and the EOS-TD is used to measure the timing change

caused by the amplitude modulation. As shown in Fig. 3b, the measured ATC coefficients [in $s (\Delta P/P)^{-1}$ unit] at 3-mA average photocurrent condition are 3.8 ps (1.7 ps) and 81 ps (84 ps) at rising-edge and falling-edge, respectively, for a 12-GHz p-i-n (MUTC) photodiode. For both simulation and measurement results, the ATC coefficients are vastly different between rising-edge and falling-edge.”

We also added a new subsection in the Methods in p. 19-20 (lines 408-419): “**Measurement of photodiode ATC coefficients.** To quantify the ATC coefficient experimentally, a separate experiment is conducted (see Fig. 3c). An AOM is placed in front of the photodiode-under-test. The timing modulation of the photodiode is monitored by the EOS-TD when the AOM is amplitude-modulated. The amplitude modulation is measured by a 1.7-MHz photodetector using a 5% tap coupler, which is located right before the photodiode. The amount of timing modulation is measured by the EOS-TD at the middle of rising- and falling-edges of a photocurrent pulse. A 12-GHz p-i-n and MUTC photodiode is used at 3-mA photocurrent, with 20-V and 8-V bias, respectively. The used modulation frequencies are 100 Hz, 200 Hz, 500 Hz, 1 kHz, 2 kHz, 5 kHz, 10 kHz, 20 kHz and 50 kHz with modulation depth of 0.034%. As shown in Fig. 3b, we found that there is negligible frequency dependence in the measured ATC coefficients over modulation frequency, which is similar to the previous APC coefficient measurement studies¹³.”

Point 3-3

I suppose the measured RIN reduction presented in fig.4a is measured on loop. As in-loop measurements usually don't allow to draw conclusions on the measured system, this should be clearly stated.

The RIN spectrum presented in Fig. 4a is an out-of-loop measurement result. We added “**by using an out-of-loop monitoring photodiode**” in page 11 (lines 222-223), and revised Fig. 1a to include this out-of-loop RIN monitoring photodiode in the schematic diagram.

Point 3-4

Page 4: a 40dB reduction of the PSD means a 20dB reduction of the timing jitter. Maybe this should be said explicitly to not mislead the reader.

We clarified the statement in page 5 (line 110) by adding “(i.e., a factor of ~100 reduction in timing jitter)”.

Point 3-5

fig1a: inspecting the arrows at the Sagnac interferometer I would conclude that there are only exiting arrows. One may wonder how it is possible that the Sagnac has only light leaving it but no inputs.

We added arrows to clarify the paths of entering optical pulses in Fig. 1a.

Point 3-6

Fig. 1b: it is impossible to see the 60as timing error on a ns time axis. The resolution obviously exists as the spectral domain data in fig. 1c shows. It might be helpful to see a magnified time domain picture that shows the 60as timing jitter.

Fig. 1b shows the photocurrent pulse shape measurement result using a 33-GHz real-time oscilloscope. Note that different waveforms are obtained for different input optical power levels and do not represent the jitter. On the other hand, Fig 1c (timing jitter PSD) is obtained by monitoring the output of the EOS-TD, which compares the timing between the optical pulse and a specific point of the electric pulse signal. Thus, jittery waveforms in the time-domain cannot be directly obtained with our EOS-TD method. We clarified the used measurement instruments in the figure caption of Fig. 1 (underlined; “measured by a 33-GHz real-time oscilloscope” and “measured by the EOS-TD” for Fig. 1b and 1c, respectively).

Point 3-7

Fig2b is a mystery to me. The caption says the pulse position is plotted against the dependence on timing, but the axis read timing jitter vs. normalized amplitude. Why is the timing jitter of the order of 20fs instead of the claimed 60as? What is the meaning of the dashed box at the lower left corner and the arrow? I have no clue what this all means and strongly recommend to rework this graph.

Following the reviewer’s suggestion, we reworked Fig. 2b to make it easier to understand. The original 90%, 70%, ... points were all for the falling-edge and that is why they had an order of ~20 fs jitter. We labeled each point as A, B, C, ... F to clearly indicate the measured points, where A is the 50% rising-edge amplitude, and B, C, ..., F are 90%, 70%, ... , 10% falling-edge amplitude, respectively. The inset for rising-edge jitter was included because it is difficult to read the accurate jitter levels using the main plot with 25-fs y-axis scale. We also revised this part more clearly recognizable.

Point 3-8

abstract: “... using low-cost and silicon-photonics-integrable ...”. I don’t think low cost is a scientific argument. This is even more so when we speak about the cost of a photodetector as compared to the cost of a frequency comb.

We removed “low-cost” in the abstract.

Point 3-9

Equation (1) on page 14: It might help to define the function \mathcal{F} , i.e, say that it is the Fourier Transform.

We added the definition of \mathcal{F} (Fourier transform) in page 19 (line 396).

Reviewers' Comments:

Reviewer #1:

Remarks to the Author:

I thank the authors for carefully considering my comments. I am satisfied that all my concerns have been adequately addressed, and I support publication in Nature Communications.

Reviewer #2:

Remarks to the Author:

In this referee's opinion, the authors have responded excellently to the detailed suggestions by all three reviewers. The paper is now highly recommended for direct publication in Nature Methods.

Reviewer #3:

Remarks to the Author:

I have read the revised version and find that it is now ready for being published.

Point-by-point response to Reviewer #1's comments

I thank the authors for carefully considering my comments. I am satisfied that all my concerns have been adequately addressed, and I support publication in Nature Communications.

We thank Reviewer #1 for constructive suggestions and comments on our manuscript throughout the review process.

Point-by-point response to Reviewer #2's comments

In this referee's opinion, the authors have responded excellently to the detailed suggestions by all three reviewers. The paper is now highly recommended for direct publication in Nature Methods.

We thank Reviewer #2 for constructive suggestions and comments on our manuscript throughout the review process.

Point-by-point response to Reviewer #3's comments

I have read the revised version and find that it is now ready for being published.

We thank Reviewer #3 for constructive suggestions and comments on our manuscript throughout the review process.